# Thermo-Mechanical Processing as Method Decreasing Delta-Ferrite and Improving the Impact Toughness of the Novel 12% Cr Steels with Low N and High B Contents

**DOI:** 10.3390/ma15248861

**Published:** 2022-12-12

**Authors:** Alexandra Fedoseeva, Anastasiia Dolzhenko, Rustam Kaibyshev

**Affiliations:** 1Laboratory of Prospective Steels for Agricultural machinery, Russian State Agrarian University—Moscow Timiryazev Agricultural Academy, Timiryazevskaya, 49, 127550 Moscow, Russia; 2Laboratory for Mechanical Properties of Nanostructured Materials and Superalloys, Belgorod National Research University, Pobeda 85, 308015 Belgorod, Russia

**Keywords:** heat-resistant martensitic steels, thermo-mechanical processing, structure, impact loads, electron microscopy

## Abstract

The universal thermo-mechanical processing including the interim long-term annealing together with forging for three 12% Cr martensitic steels with different alloying. This thermo-mechanical processing remarkably increases the impact toughness of these steels in wide temperature ranges and reduces the ductile-brittle transition temperature by 10–20 K. There is a 25 °C impact toughness of all 12% Cr steels subjected to the thermo-mechanical processing exceeds 60 J cm^−2^. Such an increment in impact toughness is accompanied with the significant changes in the structures of all 12% Cr steels with different alloying. The common feature for all 12% Cr steels subjected to the thermo-mechanical processing is found to be a noticeable decrease in delta-ferrite amount. In the steels containing Ta, the decrease in the mean size of prior austenite grains by 20–26% was revealed. For the 12% Cr steels with ultra-low N content, the thermo-mechanical processing provides the changes in the dispersion of M_23_C_6_ carbides and MX carbonitrides.

## 1. Introduction

The 12% Cr martensitic steels are considered to be potential materials for production of blades of steam turbines for fossil power plants operating at the temperatures higher than 620 °C [1,2]. Improved creep resistance of the 12% Cr steels is achieved by the formation of the tempered lath structure, which is stabilized by the dispersion of secondary phase particles [1,2,3,4,5,6,7,8,9,10]. High Cr content provides good corrosion resistance but stimulates the formation of delta-ferrite [11,12,13]. There are two main disadvantages of the 12% Cr steels. First, the creep strength breakdown appears due to the precipitation of the coarse Z-phase particles instead of the nanoscale MX carbonitrides during long-term creep [14,15,16,17,18]. The effective method to prevent the precipitation of Z-phase in 10–12% Cr steels during creep or aging is a decrease in nitrogen content to 0.01 wt.% or less [7,19,20]. This eliminates both V-rich MX and Z-phase particles [19,20]. The decrease in N content gives an opportunity to increase in B content without the formation of BN nitrides at high temperatures [2]. High B content positively effects on the dispersion of M_23_C_6_ carbides decreasing their mean size that improves the creep resistance [2]. The 12% Cr steels with such alloying shows the value of 1% creep limit of about 80 MPa that is similar with that for the P92 + 3% Co steel because of higher solid solution strengthening [19,20]. Second, this is susceptibility of the 12% Cr steels with the ultra-low N content to the formation of delta-ferrite that causes an increase in ductile-brittle transition temperature (DBTT) higher than room one [21]. Delta-ferrite amount is determined by heat treatment and/or chemical composition of the 10–12% Cr martensitic steels. The negative effect of delta-ferrite on impact toughness of these steels is found to be the precipitation of W-rich or Cr-rich carbide chains at the boundaries of delta-ferrite/martensite [11,12,21,22,23,24,25,26,27,28,29,30]. These brittle carbide chains prevent the deformation of delta-ferrite together with martensitic matrix which causes the formation of cracks. There are the ways to decrease in delta-ferrite amount via combination of long-term annealing at high temperatures and deformation [23,24,27,29]. Moreover, the M_23_C_6_ carbides also can be reason of embrittlement in a case of their precipitation as the dense continuous chains along the boundaries of martensitic laths [21,31,32]. Such a precipitation of M_23_C_6_ carbides acts as thin brittle layer that restricts the deformation transfer from one lath to other lath [21,31].

In the study [33], we reported on the positive impact of thermo-mechanical processing on the decrease in delta-ferrite to 3.8% (vs. 10% as it was after conventional heat treatment) and increase in impact toughness of the 12% Cr steel with the low N and high B contents. As it was mentioned in [33], the significant increment in entire Charpy energy curve for the 12% Cr steel via the thermo-mechanical processing was accompanied with the structural changes in (a) the delta-ferrite amount, (b) the dispersion of M_23_C_6_ carbides and (c) an additional precipitation of nanoscale MX carbonitrides. However, such thermo-mechanical processing provides a slight decrease in the DBTT from 70 °C (after conventional heat treatment) to 65 °C [33]. This led to the modification of the thermo-mechanical processing via decreasing the temperature of homogenization annealing from 1200 °C to 1150 °C as it was in a conventional heat treatment. Such a modification provides the significant improving impact toughness and shifting DBTT to the side of lower temperatures by 20 K [34]. However, the application of such a modified thermo-mechanical processing for other 12% Cr steels with different alloying is an open question. The present research is consistent development of previous works directed to improving the impact toughness of the 12% Cr steels. Thus, the aim of the present research is verification of the positive effect of modified thermo-mechanical processing for three 12% Cr steels with different alloying containing low N and high B additives.

## 2. Materials and Methods

Three 12% Cr martensitic steels were melted using vacuum-induction furnaces to obtain the low N content. The content of oxygen in all steels did not exceed 0.01 wt.%. The N and O contents were estimated using a METEK-300/600 gas analyser (METEKPROM Ltd., Izhevsk, Russia). The contents of other elements were obtained via optical emission spectrometer FOUNDRY-MASTER UVR (Oxford Instruments, Ambingdon, UK). The chemical compositions are summarized in Table 1.

These steels were subjected to the conventional heat treatment (CHT) and thermo-mechanical processing (TMP). The CHT consisted of homogenization annealing at a temperature of 1150 °C for 16 h following forging to a true strain of ~1, air cooling, normalizing at a temperature of 1050–1070 °C for 1 h, air cooling, final tempering at a temperature of 770 °C for 3 h, and air cooling again. The TMP represented in Figure 1 consisted of homogenization annealing at a temperature of 1150 °C for 16 h with following forging to a true strain of ~1, air cooling, interim annealing at a temperature of 1050 °C for 4 h with following forging at 1050 °C to a true strain of ~4, and annealing again at 1050 °C for 4 h, air cooling, normalizing at a temperature of 1000 °C for 24 h, air cooling, final tempering at a temperature of 770 °C for 3 h, and air cooling [34].

Charpy V-notch specimens with a cross section of 10 mm × 10 mm and a length of 55 mm and 2-mm V-notch were evaluated at the temperatures ranging from −40 °C to 140 °C using an Instron 450 J impact machine, Model SI-1M (Instron, Norwood, MA, USA). The DBTT was estimated in accordance with the ASTM E-23 standard [35].

An Olympus GX70 optical microscope (Olympus Corporation, Tokyo, Japan) was used for observation of delta-ferrite and prior austenite grains (PAGs). The samples for optical microscope were prepared by etching in a solution of 1% HF, 2% HNO_3_, and 97% H_2_O. The mean size of PAGs and delta-ferrite was estimated by the linear intercept method. The amount of δ-ferrite was evaluated as the relation of sum of lengths of δ-ferrite grains to the length of intercept line. The analysis of lath structure together with the secondary phase particles was conducted using transmission electron microscope (TEM), JEOL–2100 (JEOL Ltd., Tokyo, Japan), and scanning electron microscope (SEM), Quanta 600FEG ((FEI, Hillsboro, OR, USA). The foils were obtained in a solution of 10% perchloric acid and 90% acetic acid using Struers Tenupol-5 machine (Struers Inc., Cleveland, OH, USA) at a room temperature and a voltage of 23 V. The carbon replicas were prepared using vacuum deposition machine, Q 150REQuorum (Quorum Technologies, Laughton, UK). The dislocation density was evaluated as count of intersections of free dislocations with foil surfaces per unit area. Equilibrium phase diagrams were evaluated via Thermo-Calc software (Version 5.0.4 75, Thermo-Calc software AB, Stockholm, Sweden, 2010). Analysis of the mechanical properties and structural characterization technique was considered in detail in [7,8,10,18,19,20,21,31,32,33,36].

## 3. Results and Discussion

### 3.1. Impact Tests

The results of impact tests together with “load vs. deflection” curves at the different temperatures obtained for the 12% Cr steels subjected to the CHT and TMP are demonstrated in Figure 2. It should be noted that the TMP had positive influence on the resistance of impact loads for all 12% Cr steels studied and shifts the DBTT to a side of lower temperatures and higher energies. For the 12CrTaNb0.003N steel, the difference between the DBTTs corresponding to the CHT and TMP was 20 K, wherein all Charpy energy curve shifted up to ~50 J cm^−2^ (Figure 2a). For the 12CrNb0.003N steel, the DBTT decreased from 65 °C for the CHT to 55 °C for the TMP, wherein the significant increment in Charpy energy at all temperatures excepting room temperature was not found (Figure 2c,d). For the 12CrTaNb0.02N steel, the decrease in the DBTT was 15 K; all Charpy energy curve demonstrated as much as a ~30 J cm^−2^ increase (Figure 2e).

At the temperatures lower than 10 °C, the “load vs. deflection” curves represented the maximal peak, which was characterized for 100%-brittle fracture (Figure 2b,d,e). This was typical for all steels studied (Figure 2b,d,f and Figure 3). For all steels subjected to the TMP, the load, which was required for crack initiation and propagation, was higher compared with the CHT, but the crack growth occurred in the brittle manner for all steels subjected to both CHT and TMP (Figure 2b,d,f). Note that for the steels containing Ta, the impact toughness exceeds 40 J cm^−2^, even at temperatures lower than −20 °C (Figure 2a,e)

At room temperature, for the steels with ultra-low N content, the “load vs. deflection” curves for the samples subjected to the TMP significantly differed from that for the samples subjected to the CHT (Figure 2b,d). For the samples subjected to the CHT, the “load vs. deflection” curves consisted of one maximal peak, which was similar with curves for lower temperatures. On the other hand, after the TMP, the “load vs. deflection” curves obtained a plateau that indicated the crack growth in a ductile manner [37]. For the 12CrTaNb0.02N steel, both “load vs. deflection” curves contained the zone of stable crack propagation, while the fracture toughness of the samples subjected to the TMP was twice higher compared with the CHT (Figure 2f). At the room temperature, the value of impact toughness exceeded 60 J cm^−2^ for all 12% Cr martensitic steels subjected to the TMP that is feasible for commercial application as a structural material for fossil power plants [1,2,38]. At the temperatures close to the DBTT and higher, all four zones of crack initiation and propagation including stable and unstable crack growth could be distinguished for all steels exposed of both processing routes. For temperatures more than 80–100 °C, only ductile fracture propagation was revealed for all 12% Cr steels (Figure 2).

The area under the “load vs. deflection” that curves up to the maximal point is considered to be the crack initiation energy, *E_i_*, while the remaining area corresponds to the crack propagation energy, *E_p_*. These values for *E_i_* and *E_p_* are shown in Table 2 and Table 3, respectively. The maximum load on the “load vs. deflection” curves can be easily expressed in term of the stress using the following relation [32,39]:(1)$$\sigma_{M}=\frac{\beta SP}{2C_{f}{\left(W-a\right)}^{2}B}$$
where *σ_M_* is the maximum stress; *S* is the span (=40 mm); *P* is the maximum load; *W* is the specimen width (=10 mm); *B* is the specimen thickness (=10 mm); *a* is the notch depth (=2 mm); *C_f_* is the constraint factor (=1.363 for ASTM tup); *β* is a constant depending on the yielding criterion; and *β* = 2 for Tresca criterion. The calculated values for maximum stress are summarized in Table 4.

At lower temperatures for the steels containing Ta, crack initiation energy in the samples subjected to the TMP was considerably higher compared with the CHT (Table 2). The increment of crack initiation energy was 76% for the 12CrTaNb0.003N steel, and from 34% to 66% for the 12CrTaNb0.02N steel at the test temperatures lower than 10 °C. As the test temperature increased, the increment of crack initiation energy decreased in both steels containing Ta (Table 2). However, for the 12CrTaNb0.003N steel, the TMP provided higher values of crack initiation energy. Whereas for the 12CrTaNb0.02N steel, the values of crack initiation energy was close for the samples exposed of both CHT and TMP (Table 2). In contrast, at temperatures lower than 10 °C, the 12CrNb0.003N steel crack initiation energy for the samples subjected to the CHT was twice as high compared to the TMP (Table 2). As the test temperature increased, the crack initiation energy became similar for the samples subjected to both CHT and TMP, except in room temperature (Table 2). At room temperature, the increment in crack initiation energy was 62% for the samples of 12CrNb0.003N steel subjected to the TMP (Table 2). The TMP route significantly increased the crack propagation energy for all 12% Cr steels studied (Table 3). This should be noted that increment in the crack propagation energy was well-defined and comprised 3–5 times for the 12CrTaNb0.003N steel, whereas for the other 12% Cr steels, the increment in the crack propagation energy was revealed for some test temperatures, only (Table 3). For example, at room temperature, the increases in crack propagation energy were two and four times for the 12CrNb0.003N and 12CrTaNb0.02N steels, respectively (Table 3). The results of crack energy (Table 2 and Table 3) were in accordance with maximum stresses (Table 4). For the steels with ultra-low N content, the significant increment in maximum stress was observed for the temperatures ranging from −20 °C to 25 °C (Table 4). In contrast, for the 12CrTaNb0.02N steel, the TMP provided the increase in maximum stress at lower temperatures only (Table 4).

Thus, the TMP route positively affects the impact toughness of the different 12% Cr steels in wide range of test temperatures providing the high values of energies of both crack initiation and propagation (Figure 2, Table 2 and Table 3).

### 3.2. Fractography after Impact Tests

The TMP had a significant effect on −20 °C impact toughness for the 12% Cr steels containing Ta via an increase in the crack initiation energy (Figure 2a,e and Table 2). Figure 3 demonstrates the fractured surfaces in the zone of unstable crack propagation for the samples of the different 12% Cr steels after −20 °C impact test. For the 12CrTaNb0.003N steel (Figure 3a,d), the 110- and 200-μm crack initiation zone was observed for the CHT and TMP, respectively. This correlated with the higher crack initiation energy after TMP (Table 2). The unstable crack propagation zone contained cleavage facets and separated tear ridges exhibiting dimples for both CHT and TMP (Figure 3a,d). After TMP, the fraction of tear ridges was twice higher that corresponded to higher crack propagation energy (Table 3). For the 12CrTaNb0.02N steel (Figure 3c,f), the about 160- and 200-μm crack initiation zone was exhibited for the CHT and TMP, respectively, and a longer crack initiation zone corresponded to higher crack initiation energy (Table 2). However, no evidence for the tear ridges in the zone of unstable crack propagation was observed for the samples exposed of the TMP (Figure 3f). In contrast, the sample exposed of the CHT demonstrated the presence of the tear ridges with dimples (Figure 3c). This fact is in accordance with results of crack propagation energy (Table 3). After the TMP, the energy of crack propagation was less compared with the CHT (Table 3). For the 12CrNb0.003N steel (Figure 3b,e), the about 95-μm crack initiation zone was observed for both the CHT and TMP that indicated low crack initiation energy (Table 2). The unstable crack propagation zone contained 100% cleavage facets without the presence of tear ridges after both the CHT and TMP routes (Figure 3b,e) that provided the low values of crack propagation energy (Table 3). It should be noted that the 12CrNb0.003N steel demonstrated the lowest value of −20 °C impact toughness (Figure 2b) among all 12% Cr steels studied.

Moreover, the TMP positively affects 25 °C impact toughness for the all 12% Cr steels studied, increasing both crack initiation and propagation energy (Figure 2 and Table 2 and Table 3). Figure 4 demonstrates the fractured surfaces in the zones of stable and unstable crack propagation for the samples of the 12CrTaNb0.003N steel exposed of the CHT and TMP after 25 °C impact test. Other 12% Cr studied steels had the same fractography images. The principal difference of fractured surface of the samples of the different 12% Cr steels subjected to the TMP was the presence of the zone of stable crack propagation with a length ranging from 0.73 mm for the 12CrNb0.003N steel to 1.04 mm for the 12CrTaNb0.02N steel and well-defined arrested zone (Figure 4d–f and Table 5). Both of these zones corresponded to ductile fracture mechanism. High crack initiation energy of the samples subjected to the TMP was caused by the appearance of ductile fracture components (Figure 4d–f and Table 2), wherein the length of crack initiation zone had similar values for all 12% Cr steels studied (Table 5). Moreover, even the unstable crack propagation zone contained the high fraction of tear ridges with dimples (Figure 4c,f) that provided the increase in the crack propagation energy compared with the CHT (Table 3). Shear fracture estimated in accordance with the ASTM E-23 standard [35] exceeded 30% for all 12% Cr steels studied.

### 3.3. Phase Equilibrium Modeling

Equilibrium phase diagrams for the different 12% Cr steels via Thermo-Calc software are represented in Figure 5. The phase transition temperatures of *A*_1_ (start of α-ferrite → austenite transition), *A*_3_ (finish of α-ferrite → austenite transition), and *A*_4_ (start of austenite → δ-ferrite transition) for the different 12% Cr steels are summarized in Table 6. Alloying of the 12% Cr steels via a change in N and Ta/Nb contents strongly determines the phase diagram and temperatures of phase transitions (Figure 5, Table 6). For all 12% Cr steels, the *A*_1_ temperatures were close to 840 °C (Table 6). In the steels with ultra-low N content, the phase compositions up to A_4_ temperatures were similar excepting the formation TaX phase in the steel containing Ta and NbX phase in the steel containing Nb (Figure 5). In the 12CrTaNb0.02N steel, the Z-phase was stable up to 810 °C. The significance difference between the 12% Cr steels was the temperature of the delta-ferrite formation (Table 6). For the 12CrNb0.003N steel, the *A*_4_ temperature was higher on 50–80 K compared with the 12% Cr steels containing Ta (Table 6). Moreover, for the 12CrNb0.003N steel, no area where austenite and delta-ferrite co-existed was revealed (Figure 5b). For the 12% Cr steels containing Ta, the A_4_ temperature was close to the interim annealing and forging in the TMP route.

### 3.4. Structure of the 12CrTaNb0.003N Steel Subjected to CHT and TMP

The microstructures and the secondary phase particles that developed in the 12CrTaNb0.003N steel after the CHT and TMP are shown in Figure 6 and Figure 7. The parameters of structure are summarized in Table 7.

The TMP route had strong influence on the fraction, size of delta-ferrite grains, and the mean size of PAGs (Figure 6a,d and Table 7). The fraction of delta-ferrite decreased from 10 ± 2% after CHT to 1 ± 0.2%, and after TMP (Figure 6a,d and Table 7); the size of delta-ferrite grains reduced to 7 ± 2 μm (Figure 6a,d). The mean size of PAGs reduced by 20% (Table 7). Large white particles observed in Z-contrast SEM (Figure 6b,e) were enriched in W, and were considered to be W_2_B in accordance with phase diagram represented in Figure 5a. These particles were unable to dissolve at the temperature of homogenization annealing (Figure 5a) and were observed after both the CHT and TMP (Figure 6b,d, Table 7).

After both the CHT and TMP, the tempered lath structure, with a mean size of (0.28–0.29) ± 0.05 μm and high dislocation density of (2.0–2.2) × 10^14^ m^−2^, was revealed (Figure 6c,f, Table 7). M_23_C_6_ carbide was found to be main secondary phase with a volume fraction of 2.25% for both regimens (Figure 7a,c, Table 7). This phase decorated the boundaries of all structural elements of the tempered lath structure (Figure 6c,f). Their average chemical composition was (70–72)% Cr, 25% Fe, and (3–5)% W. However, after the TMP, the principal changes in the dispersion of M_23_C_6_ carbides were founded. First, the mean size of M_23_C_6_ carbides was +30% larger compared with the CHT (Table 7). Second, the particle number density estimated as particle count per unit of length of martensitic lath boundary decreased from 2.5 μm^−1^ after the CHT to 1.5 μm^−1^ after the TMP.

(Ta,Nb)X carbonitrides were randomly distributed in the bulk of the lath structure after both the CHT and TMP routes (Figure 7). The volume fraction of this phase estimated via Thermo-Calc software was 0.08% after both regimens (Table 7). The TMP led to +90% increase in the mean size of (Ta,Nb)X particles. This was related to the long exposure in austenitic region during the TMP route. The chemical composition of (Ta,Nb)X after the CHT and TMP was similar and was (75–80)% Ta, (7–25)% Nb, and (0–13)% Cr + Fe (Figure 7c). As it was shown in Ref. [33], additional non-equilibrium V-rich MX phase with a mean size of 30 ± 5 nm precipitated after the TMP (Figure 7d, Table 6). The average chemical composition of this phase was 75% V, 20% Nb, and 5% Cr (Figure 7d). Since this phase was non-equilibrium and could not be estimated via Thermo-Calc software, their volume fraction was calculated using TEM images in dark field [33] as follows:(2)$$F_{v}=\frac{4\pi }{3}\frac{N_{T}r^{3}}{A_{s}\left(\tau +2r\right)}$$
where *N_T_* is particle count (−), *r* is a particle radius (nm), *A_s_* is square of TEM image (nm^2^), and *τ* is thickness of carbon replica (10 nm). The fraction of VX carbonitrides was 0.04%.

Thus, the TMP changes the structure of the 12CrTaNb0.003 steel as follows:(a)a decrease in the fraction of delta-ferrite to 1%;(b)a decrease in the mean size of PAGs to 40 μm;(c)an increase in the mean size of M_23_C_6_ carbides up to 65 nm together with a decrease in the particle number density to 1.5 μm^−1^;(d)an additional precipitation of VX carbonitrides with the mean size of 30 nm and volume fraction of 0.04%.

### 3.5. Structure of the 12CrNb0.003N Steel Subjected to CHT and TMP

The microstructures and the secondary phase particles that developed in the 12CrNb0.003N steel after the CHT and TMP are shown in Figure 8 and Figure 9. The parameters of structure are summarized in Table 8.

The significant reduction in the delta-ferrite amount to the almost full disappearance together with reduction of its mean size was observed in the 12CrNb0.003N steel subjected to the TMP (Figure 8a,d and Table 8). The mean size of PAGs retained about 50 ± 4 μm after both routes (Table 8). Coarse W_2_B particles with sizes more than 600 ± 50 nm observed after both routes (Figure 8b,e, Table 8). It is worth noting that a decrease in the fraction of delta-ferrite was the most remarkable structural change in the 12CrNb0.003N steel after the TMP (Table 8).

The tempered martensite lath structure was formed after both CHT and TMP regimens (Figure 8c,f). The mean widths of martensitic laths and values of dislocation density were comparable after the CHT and TMP (Table 8). There was no evidence for the changes of M_23_C_6_ carbides and NbX carbonitrides was revealed (Figure 9). The mean size, volume fraction, and chemical composition of these phases were nearly the same after the CHT and TMP (Table 8). The particle number density of M_23_C_6_ carbides was about 2.3 μm^−1^ after both routes. The additional precipitation of VX carbonitrides was also distinguished in the 12CrNb0.003N steel exposed of the TMP (Figure 9d). The chemical composition of VX phase was 37% V, 20% Nb, 39% Cr, and 4% Fe. The mean size was 12 ± 5 nm (Figure 9d and Table 8). The volume fraction was also estimated using Equation (2) and comprised 0.03%. These features strongly differed from those in the 12CrTaNb0.003N steel. This can indicate that VX particles in the 12CrTaNb0.003N steel precipitated in austenitic region, whereas the formation of VX particles in the 12CrNb0.003N steel occurred in ferritic region [33].

Thus, the TMP changes the structure of the 12CrNb0.003 steel as follows:(a)a decrease in the delta-ferrite amounted to an almost full disappearance;(b)an additional precipitation of VX carbonitrides with the mean size of 12 nm and volume fraction of 0.03%.

### 3.6. Structure of the 12CrTaNb0.02N Steel Subjected to CHT and TMP

The microstructures and the secondary phase particles that developed in the 12CrTaNb0.02N steel exposed to the CHT and TMP are shown in Figure 10 and Figure 11. The parameters of structure are summarized in Table 9.

The TMP provided the decrease in the fraction of delta-ferrite from 15% as it was after the CHT to 10% (Figure 10a,d and Table 9). The mean size of the delta-ferrite decreased from 38 μm after the CHT to 19 μm after the TMP; additionally, the delta-ferrite grains tended to form a polygonal shape (Figure 10a,d). Such an insignificant decrease in the delta-ferrite amount was caused by the fact that A_4_ temperature of the formation of delta-ferrite coincided with interim annealing with forging (Figure 1 and Figure 5c, Table 6) that restricted the dissolution of this phase. The TMP provided remarkable reduction of the mean size of PAGs from 58 ± 4 μm as it was after the CHT to 43 ± 4 μm (Table 9). The temperature of homogenization annealing was not high enough to dissolve the coarse W_2_B particles, and they had the mean size more than 700 ± 50 nm after both routes (Figure 10b,e and Table 9). The TMP did not lead to the decrease their amount (Figure 10e).

In the 12CrTaNb0.02N steel, the formation of tempered lath structure was observed after both CHT and TMP regimens (Figure 10c,f). Both CHT and TMP provided the similar values of martensitic laths of 0.35 ± 0.05 μm and high dislocation density (Table 9). M_23_C_6_ carbides located at the martensitic laths had relatively large size of 64 ± 5 nm and low particle number density of 1.4 μm^−1^ even after the CHT (Figure 11a and Table 9). The TMP did not lead to the significant changes in the dispersion of M_23_C_6_ carbides (Figure 11a,d and Table 9). The effect of the TMP on the dispersion of (Ta,Nb)X carbonitrides was also not revealed. Similar mean size of this phase of 80 ± 5 nm and volume fraction of 0.08% was estimated after both routes (Table 9). In contrast with other 12% Cr steels with the ultra-low N content, VX phase could be considered as quasi-equilibrium phase, when Z-phase was suspended, and could be predicted using Thermo-Calc software (Table 9). Additionally, the volume fraction of VX phase was estimated via Equation (1) using dark field in TEM of carbon replicas (Figure 11c,f). The last method provided the volume fraction of VX carbonitrides of 0.08% regardless of the treatment. The mean size of VX phase comprised about 30 ± 5 nm after both routes (Figure 11c,f and Table 9).

Thus, the TMP changes the structure of the 12CrTaNb0.02 steel as follows:(a)a decrease in the delta-ferrite amount to 10%;(b)a decrease in the mean size of PAGs to 43 μm.

### 3.7. The Relationship between the Microstructural Characteristics and Impact Toughness

The TMP improves the fracture toughness of the 12% Cr steels with the low N and high B contents and shifts the DBTT to a side of lower temperatures (Figure 2). This effect is caused by strong influence of the TMP on the structure of such steels. The role of embrittlement agents for the 12% Cr steels can play (a) delta-ferrite, (b) coarse W-rich particles, and (c) high M_23_C_6_ carbide density in laths, packets, blocks, PAGs [11,12,22,23,24,25,26,27,28,29,30,32]. Thus, the improved fracture toughness is achieved at the formation of 100% tempered lath structure without delta-ferrite grains with relatively small size of PAGs [22,23,24,25,26,27,28,29,30]. The boundaries of martensitic laths should be decorated by relatively coarse M_23_C_6_ rarely distributed along the boundary lines to prevent the formation of thin brittle layers [21,31,33]. The precipitation of fine MX with high volume fraction is favourable. The coarse W-rich particles should be dissolved. The structure of the 12CrTaNb0.003 steel exposed of the TMP was maximal close to the “ideal” situation (Figure 6 and Figure 7) that provided the decrease in DBTT by 20 K (Figure 2a). Moreover, the increase in the entire Charpy energy curve, including the lower and upper shelf energies, was also accompanied with the reducing the mean size of PAGs and changes in a dispersion of the secondary phase particles (Figure 2a, Figure 6 and Figure 7c,d). The amount of the large W_2_B particles can be reduced by an increase in homogenization annealing as it was shown in Ref. [33]. However, this causes the formation of a large amount of delta-ferrite, which is retained even after high-temperature long-term annealing and forging with a true strain of 5 [33]. Moreover, it seems that the delta-ferrite has more negative effect on the impact toughness and DBTT than the coarse W_2_B particles [33].

In the 12CrNb0.003N steel, the almost 100% tempered lath structure occurs after the TMP (Figure 8). The almost full elimination of delta-ferrite leads to the reducing the DBTT by 10 K (Figure 2b and Figure 8c). However, no increment in the energies of upper and lower shelf regions via the TMP was revealed (Figure 2b). This can be caused by the fact that no evidence for the changes in the mean size of PAGs and/or a dispersion of M_23_C_6_ carbides was observed, while fine VX carbonitrides were additionally precipitated, but their volume fraction was insignificant (Figure 9). The further refinement of the PAGs is restricted since most of the NbX and VX particles precipitate in ferritic region. To change the dispersion of M_23_C_6_ carbides is possible only by the change in the tempering conditions.

The structure of the 12CrTaNb0.02N steel contained about 10% of delta-ferrite after the TMP (Figure 10). A decrease in the delta-ferrite fraction from 10–15% to 0–10% in the 12% Cr steels provides the decrease in the DBTT by 10–20 K (Figure 2) [19,31], whereas in a 13Cr-4Ni steel containing 0.03% C and Ti, the reduction of delta-ferrite content from about 8% to 0% leads to remarkable decrease in the DBTT by 75 K [11]. It seems that amount of delta-ferrite determines the DBTT position, whereas the size of PAGs and/or the dispersion of the secondary phase particles affect the energies of upper and lower shelf regions. Even after CHT, the dispersion of M_23_C_6_ carbides in the 12CrTaNb0.02N steel corresponds to “ideal” structure, whereas MX carbonitrides are found to be too large to improve the impact toughness. For the 12CrTaNb0.02N steel, it is necessary to change the conditions of interim annealing and forging through a decrease in temperature to 1000 °C or less to provide non-equilibrium state of delta-ferrite (Figure 5c). This helps to decrease the delta-ferrite content and/or the mean size of (Ta,Nb)X carbonitrides and to improve DBTT.

However, the structure of the 12% Cr steels described above as “ideal” for good impact toughness will have the low creep resistance due to absence of threshold back-stresses [40]. It is known that good creep resistance is accompanied with the large mean size of PAGs and relatively fine M_23_C_6_ carbides densely located along lath boundaries [41]. The balance between creep resistance and impact toughness should be provided by compromise in the size of PAGs and dispersion M_23_C_6_ carbides in a case of full elimination of delta-ferrite grains. For the 12% Cr steels studied, such compromise is considered to be as (a) the mean size of PAGs not less 40–45 µm and (b) the mean size of M_23_C_6_ carbides not more 65 ± 5 nm and their particle number density along the martensitic laths not more 1.5 µm^−1^. The future investigations will be directed to checking the creep properties of the 12% Cr martensitic steels exposed of the TMP route represented in Figure 1, and to further improving the TMP by modification of interim annealing and forging with an aim to obtain more universal processing for the different 12% Cr martensitic steels.

## 4. Conclusions

The effect of the modified TMP on the structure and fracture toughness of the different 12% Cr steels with the low N and high B contents was studied. The main results can be shown as follows:The TMP positively affects the impact toughness of all different 12% Cr steels. This reduces the DBTT by 10–20 K for all 12% Cr steels. For the 12% Cr steels containing Ta, the TMP shifts all entire Charpy energy curves to higher values by 30–50 J cm^−2^, including the lower and upper shelf energies.At room temperature, the Charpy energy of all 12% Cr steels exposed to the TMP comprises of more than 60 J cm^−2^, which is feasible for commercial application as a structural material for fossil power plants. Moreover, the 12% Cr steels containing Ta retain 40 J cm^−2^ Charpy energy even at the temperatures lower than −20 °C.The TMP provides the almost full dissolution of delta-ferrite in the 12% Cr steel with ultra-low N content and decreased the fraction of delta-ferrite (about 10%) in the 12CrTaNb0.02N steel.In the 12% Cr steels containing Ta, the TMP leads to a decrease in the mean size of PAGs by 20–26%. For the 12% Cr steels with ultra-low N content, the TMP causes the changes in the dispersion of M_23_C_6_ carbides and MX carbonitrides.

## Figures and Tables

**Figure 1 materials-15-08861-f001:**
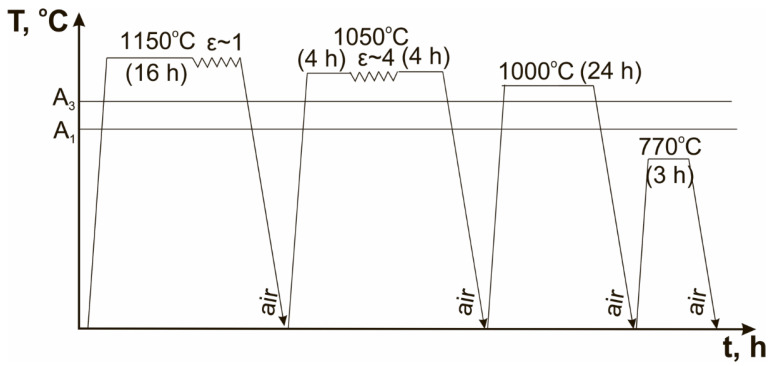
Schema of the TMP [34].

**Figure 2 materials-15-08861-f002:**
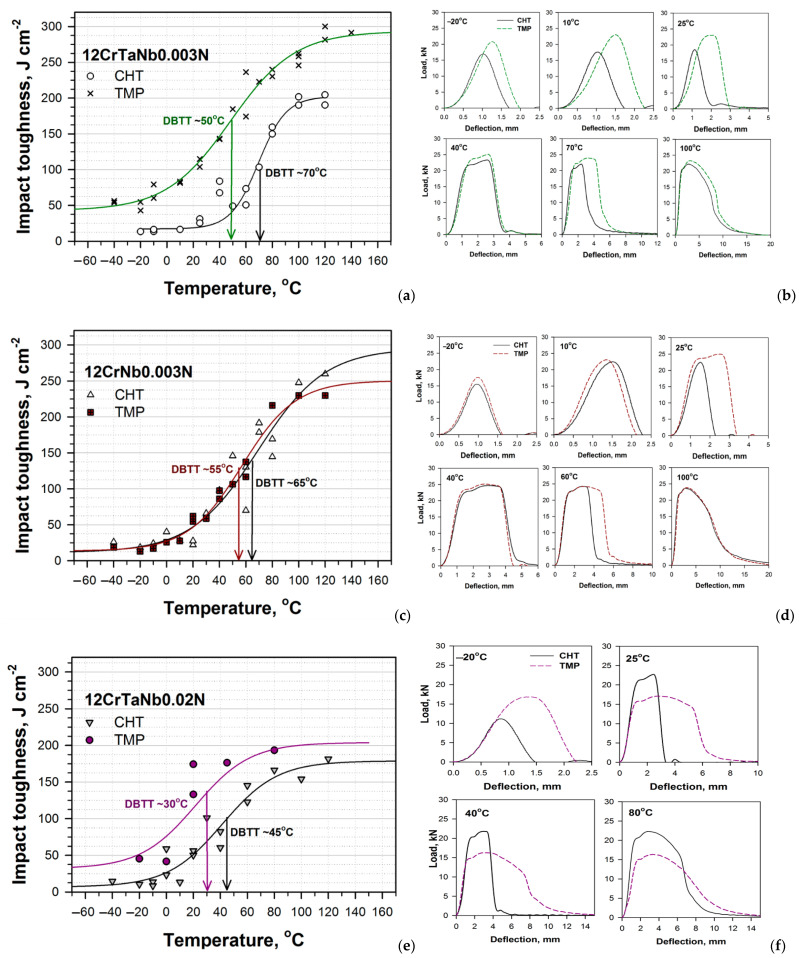
The temperature dependence of Charpy energy (**a**,**c**,**e**) together with “load vs. deflection” curves at different test temperatures (**b**,**d**,**f**) for the 12CrTaNb0.003N (**a**,**b**), 12CrNb0.003N (**c**,**d**), and 12CrTaNb0.02N (**e**,**f**) steels subjected to the CHT and TMP. DBTT means ductile-brittle transition temperature.

**Figure 3 materials-15-08861-f003:**
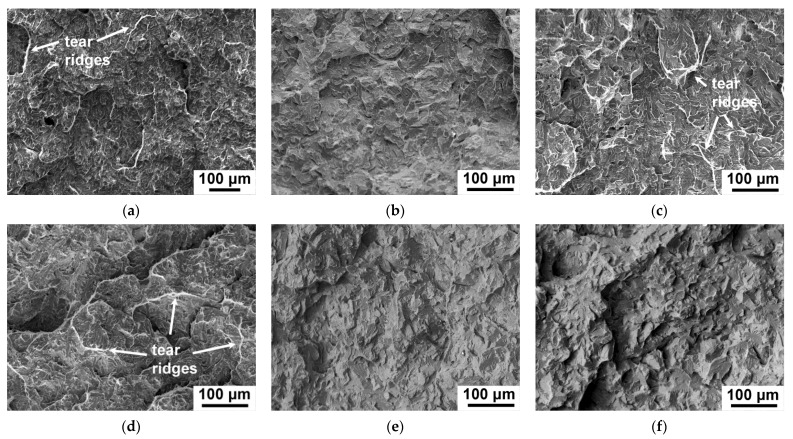
Fractography of the fractured surfaces in the unstable crack propagation zone of the 12CrTaNb0.003N (**a**,**d**), 12CrNb0.003 (**b**,**e**), and 12CrTaNb0.02 (**c**,**f**) steels subjected to the CHT (**a**–**c**) and TMP (**d**–**f**) after the −20 °C impact test.

**Figure 4 materials-15-08861-f004:**
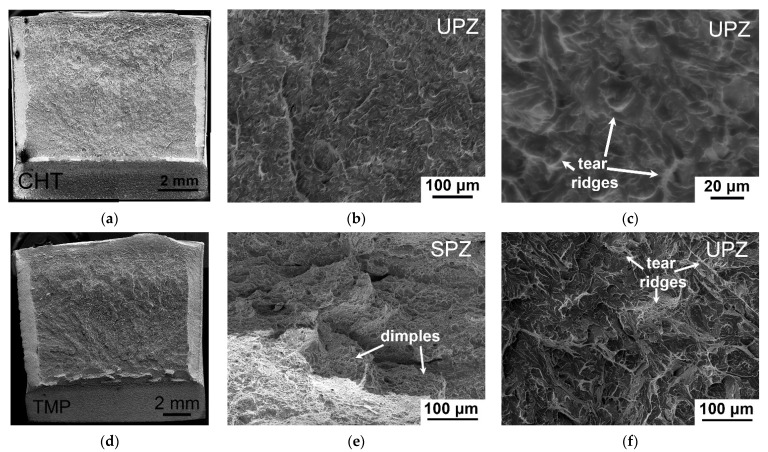
Fractography of the fractured surfaces in the stable (SPZ) and unstable crack propagation zones (UPZ) of the samples of the 12CrTaNb0.003N steel subjected to the CHT (**a**–**c**) and TMP (**d**–**f**) after the 25 °C impact test.

**Figure 5 materials-15-08861-f005:**
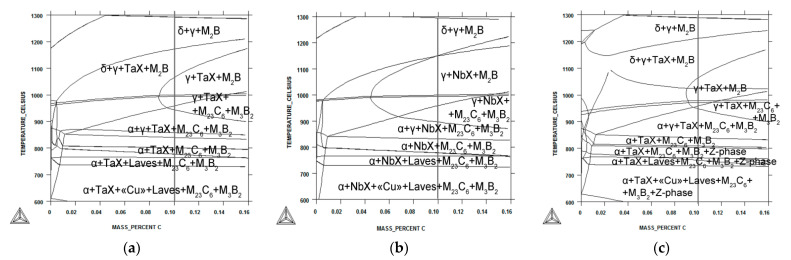
Phase diagram via Thermo-Calc software for the 12CrTaNb0.003N (**a**), 12CrNb0.003N (**b**), and 12CrTaNb0.02N (**c**) steels.

**Figure 6 materials-15-08861-f006:**
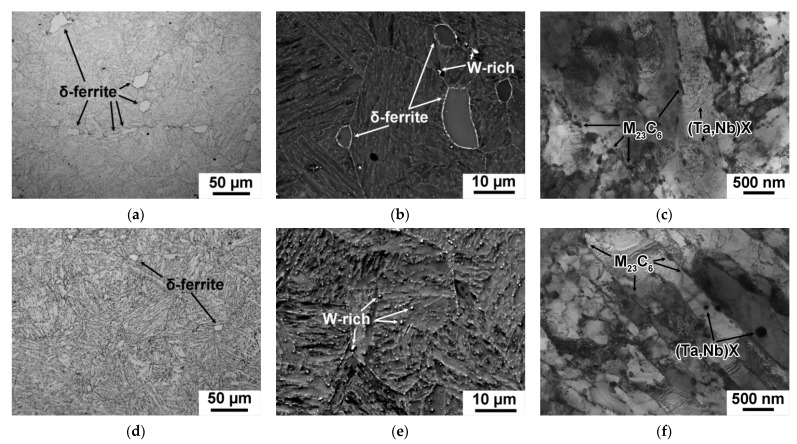
Typical microstructures in the 12CrTaNb0.003N steel subjected to the CHT (**a**–**c**) and TMP (**d**–**f**) obtained by optical microscope (**a**,**d**), SEM (**b**,**e**), and TEM (**c**,**f**).

**Figure 7 materials-15-08861-f007:**
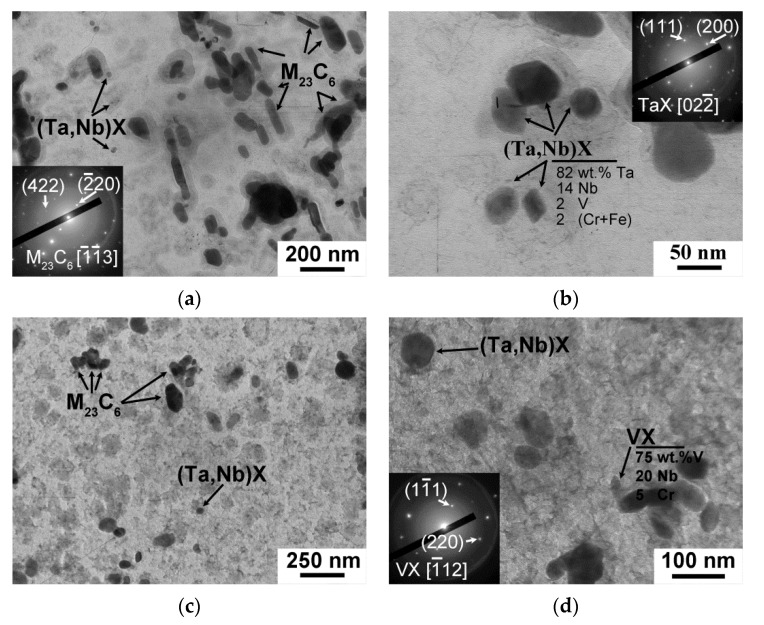
The secondary phase particles in the 12CrTaNb0.003N steel subjected to the CHT (**a**,**b**) and TMP (**c**,**d**) obtained using TEM of carbon replicas.

**Figure 8 materials-15-08861-f008:**
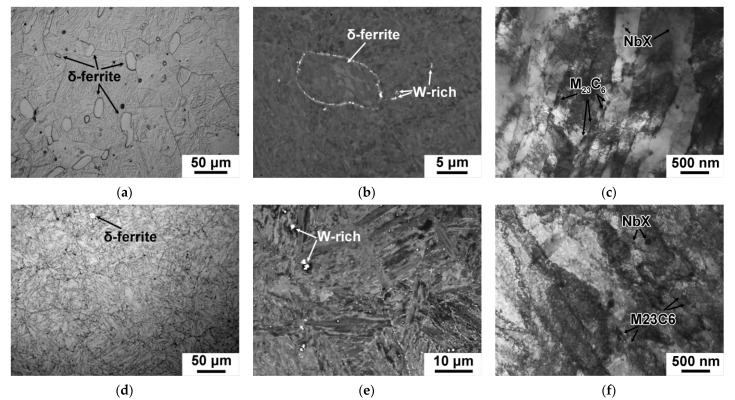
Typical microstructures in the 12CrNb0.003N steel subjected to the CHT (**a**–**c**) and TMP (**d**–**f**) obtained by optical microscope (**a**,**d**), SEM (**b**,**e**), and TEM (**c**,**f**).

**Figure 9 materials-15-08861-f009:**
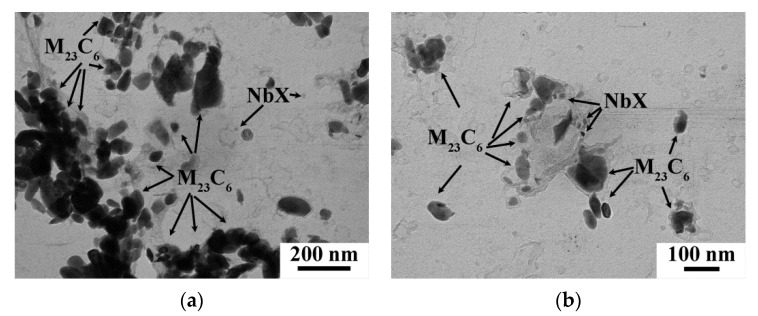
TEM images of the secondary phase particles in the 12CrNb0.003N steel subjected to the CHT (**a**,**b**) and TMP (**c**,**d**) obtained using carbon replicas.

**Figure 10 materials-15-08861-f010:**
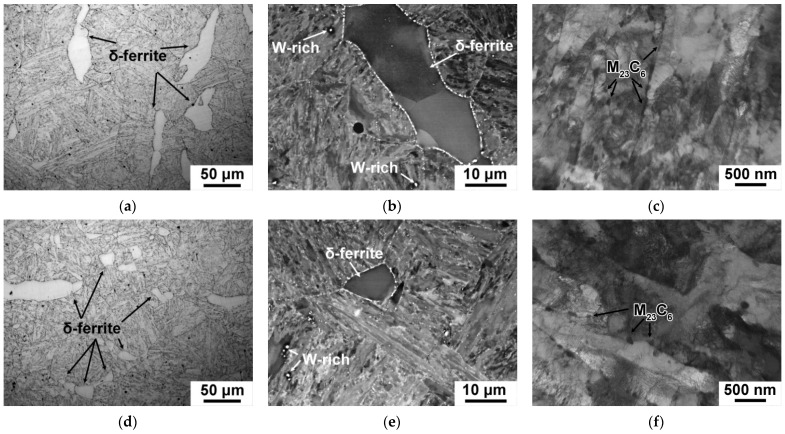
Typical microstructures in the 12CrTaNb0.02N steel subjected to the CHT (**a**–**c**) and TMP (**d**–**f**) obtained by optical metallography (**a**,**d**), SEM (**b**,**e**), and TEM (**c**,**f**).

**Figure 11 materials-15-08861-f011:**
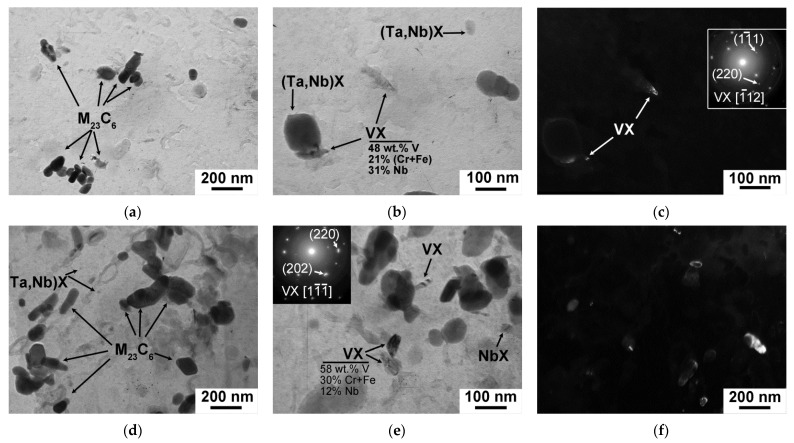
TEM images of the secondary phase particles in the 12CrTaNb0.02N steel subjected to the CHT (**a**,**b**) and TMP (**d**,**e**) obtained using carbon replicas. Dark fields in (**c**,**f**) was obtained in the reflex of the (220) [$\bar{1}$12] and (220) [1$\bar{1}\bar{1}$] VX phase, respectively.

**Table 1 materials-15-08861-t001:** The chemical compositions of the 12% Cr steels studied (in wt.%).

Ingots	Fe	C	Cr	Co	Mo	W	V	Nb	Ta	Cu	B	N
12CrTaNb0.003N	bal	0.1	11.4	3.0	0.6	2.5	0.2	0.04	0.07	0.8	0.010	0.003
12CrNb0.003N	bal	0.1	11.3	3.9	0.6	2.4	0.2	0.07	-	0.8	0.010	0.003
12CrTaNb0.02N	bal	0.1	11.9	3.0	0.6	2.2	0.3	0.05	0.07	0.8	0.012	0.02

**Table 2 materials-15-08861-t002:** Crack initiation energies *E_i_* (in J) for the different 12% Cr steels.

Test Temperature, °C	12CrTaNb0.003N		12CrNb0.003N		12CrTaNb0.02N	
	CHT	TMP	CHT	TMP	CHT	TMP
−20	6.3	26.3	15.7	7.2	8.2	12.4
0	6.2	26.3	16.0	6.9	6.9	20.1
10	7.7	31.5	16.1	15.2	-	-
25	8.9	51.4	16.2	43.1	37.0	35.3
40	40.1	50.1	49.2	46.9	41.6	37.9
60	36.4	61.9	48.6	47.7	40.6	-
80	42.6	57.8	47.1	48.9	41.8	38.0
100	44.6	50.2	45.5	46.7	41.5	-
120	45.0	50.1	47.1	46.6	39.6	-

**Table 3 materials-15-08861-t003:** Crack propagation energies *E_i_* (in J) for the different 12% Cr steels.

Test Temperature, °C	12CrTaNb0.003N		12CrNb0.003N		12CrTaNb0.02N	
	CHT	TMP	CHT	TMP	CHT	TMP
−20	5.3	11.0	3.9	4.9	13.8	8.2
0	5.5	9.7	7.1	5.3	4.5	10.2
10	5.6	11.5	6.4	10.2	-	-
25	11.5	17.6	6.5	12.1	12.3	52.9
40	14.1	38.1	35.5	33.7	25.1	76.7
60	22.4	118.7	29.2	64.2	83.3	-
80	76.3	110.4	87.0	129.6	94.5	80.0
100	107.9	133.8	153.0	148.0	80.4	-
120	122.1	169.1	161.4	147.1	113.7	-

**Table 4 materials-15-08861-t004:** The maximum stress (in MPa) for the different 12% Cr steels.

Test Temperature, °C	12CrTaNb0.003N		12CrNb0.003N		12CrTaNb0.02N	
	CHT	TMP	CHT	TMP	CHT	TMP
−20	775.4	970.7	757.5	805.2	511.7	774.0
0	780.2	975.5	1071.6	763.5	732.7	764.4
10	806.1	1082.2	1030.4	1058.8	-	-
25	852.9	1098.7	1030.4	1147.7	1041.8	786.9
40	1059.2	1081.3	1131.7	1152.8	1001.0	747.0
60	1008.8	1071.6	1113.4	1110.6	1088.6	-
80	1033.6	1052.8	1102.8	1120.2	1023.9	750.2
100	1022.6	1034.0	1084.0	1095.0	954.7	-
120	1004.2	1028.1	1070.2	1076.7	1055.1	-

**Table 5 materials-15-08861-t005:** The length (in mm) of crack initiation zone (IZ), stable propagation zone (SPZ), unstable propagation zone (UPZ), and arrested zone (AZ) for the different 12% Cr steels after 25 °C impact test.

	12CrTaNb0.003N		12CrNb0.003N		12CrTaNb0.02N	
	CHT	TMP	CHT	TMP	CHT	TMP
IZ	0.26	0.24	0.20	0.24	0.27	0.26
SPZ	-	0.93	-	0.73	0.32	1.04
UPZ	7.23	6.66	7.69	6.45	6.94	4.25
AZ	0.30	-	-	0.53	1.0	2.2

**Table 6 materials-15-08861-t006:** The temperatures of equilibrium phase transformation for the 12% Cr steels studied.

Temperature, °C	12CrTaNb0.003N	12CrNb0.003N	12CrTaNb0.02N
*A* _1_	850	840	840
*A* _3_	920	900	960
*A* _4_	1080	1130	1050

**Table 7 materials-15-08861-t007:** Structural parameters of the 12CrTaNb0.003N steel exposed of the CHT and TMP.

Treatment		Structure			Particles					
	PAG * Size, μm	δ-Ferrite Fraction, %	Lath Width, μm	*ρ_disl_*, ×10^14^ m^−2^	Size, nm				Volume Fraction (by Thermo-Calc), %	
					M_23_C_6_	TaX	VX	W-Rich	M_23_C_6_	TaX
CHT	48 ± 4	10 ± 2	0.29 ± 0.05	2.0 ± 0.1	50 ± 5	50 ± 5	-	600 ± 50	2.25	0.08
TMP	40 ± 4	1 ± 0.2	0.28 ± 0.05	2.2 ± 0.1	64 ± 5	89 ± 5	30 ± 5	650 ± 50	2.25	0.08

* PAG means prior austenite grain.

**Table 8 materials-15-08861-t008:** Structural parameters of the 12CrNb0.003N steel after the CHT and TMP.

Treatment		Structure			Particles					
	PAG * size, μm	δ-Ferrite Fraction, %	Lath Width, μm	*ρ_disl_*, ×10^14^ m^−2^	Size, nm				Volume Fraction (by Thermo-Calc), %	
					M_23_C_6_	NbX	VX	W-Rich	M_23_C_6_	NbX
CHT	51 ± 4	6 ± 2	0.31 ± 0.05	1.5 ± 0.1	51 ± 5	29 ± 5	-	700 ± 50	1.60	0.08
TMP	50 ± 4	0.5 ± 0.2	0.27 ± 0.05	2.1 ± 0.1	54 ± 5	20 ± 5	12 ± 5	600 ± 50	1.60	0.08

* PAG means prior austenite grain.

**Table 9 materials-15-08861-t009:** Structural parameters of the 12CrTaNb0.02N steel.

Treatment		Structure			Particles						
	PAG * Size, μm	δ-Ferrite Fraction, %	Lath Width, μm	*ρ_disl_*, ×10^14^ m^−2^	Size, nm				Volume Fraction (by Thermo-Calc), %		
					M_23_C_6_	TaX	VX	W-Rich	M_23_C_6_	TaX	VX
CHT	58 ± 4	15 ± 2	0.35 ± 0.05	1.4 ± 0.1	64 ± 5	78 ± 5	26 ± 5	700 ± 50	1.77	0.08	0.21
TMP	43 ± 4	10 ± 2	0.40 ± 0.05	1.9 ± 0.1	68 ± 5	80 ± 5	34 ± 5	800 ± 50	1.77	0.08	0.21

* PAG means prior austenite grains.

## Data Availability

The data presented in this study are available on request from the corresponding author. The data are not publicly available because the data also forms part of an ongoing study.

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
