# Peer review of "Thermo-Mechanical Processing as Method Decreasing Delta-Ferrite and Improving the Impact Toughness of the Novel 12% Cr Steels with Low N and High B Contents"

_materials, 2022, doi:10.3390/ma15248861_

Round 1
Reviewer 1 Report
This manuscript provided a unique Thermo-mechanical processing to decrease delta-ferrite and to improve the impact toughness of a modified 12% Сr steels applied in fossil power plants. Some significant information can be found in the manuscript, in particular those for the material with low N content(0.003%). But some other properties should be presented in the paper before it is published as a strong scientific literature for the special material.
1) In the Section Materials and method, how to obtain the three type ingots should be presented more clearly. In particular, how to obtain the very low N content should be introduced at least in brief.
2)In table 1, the O content for the three materials should be presented since it is also very key for the materials.
3)Although the "load vs. deflection” curves at different temperatures" are also provided in Fig.2, but the reviewer can not see clearly what is the corresponding strength at different temperature. As we know, the impact toughness is closely related to strength, i.e the impact toughness increase is not so significant if it is with large strength decrease. Hence, it is suggested that the strength data should be summarized more clearly like table 2 or 3.
4) The grain size for the three materials after CHT and TMP should be replenished.
5)The creep properties are the most important properties for the materials, thus at least the creep strengths at high temperature after CHT and TMP should be replenished in the paper. In that case, readers can evaluatthe effectiveness of composition modification and the new heat treatment method.
Author Response
Point 1: In the Section Materials and method, how to obtain the three type ingots should be presented more clearly. In particular, how to obtain the very low N content should be introduced at least in brief.
Response 1: The text was added to the Section Materials and method: “Three 12% Cr martensitic steels with the chemical compositions summarized in Table 1 were melted using vacuum-induction furnaces to obtain the low N content.” (p. 2)
Point 2: In table 1, the O content for the three materials should be presented since it is also very key for the materials.
Response 2: The text was added to the Section Materials and method: “The content of oxygen in all steels didn’t exceed 0.01 wt.%. The N and O contents were obtained using a METEK-300/600 gas analyzer (METEKPROM Ltd., Izhevsk, Russia). The contents of other elements were obtained via optical emission spectrometer FOUNDRY-MASTER UVR (Oxford Instruments, Ambingdon, UK).” (p. 2)
Point 3: Although the "load vs. deflection” curves at different temperatures" are also provided in Fig.2, but the reviewer can not see clearly what is the corresponding strength at different temperature. As we know, the impact toughness is closely related to strength, i.e the impact toughness increase is not so significant if it is with large strength decrease. Hence, it is suggested that the strength data should be summarized more clearly like table 2 or 3.
Response 3: Table 4 with comments was added. (p. 6): “The maximum load on the “load vs. deflection” curves can be easily expressed in terms of the stress using the following relation [30,38]:
see attached file (1)
where σM is the maximum stress; S is the span (=40 mm); P is the maximum load; W is the specimen width (=10 mm); B is the specimen thickness (=10 mm); a is the notch depth (=2 mm); Cf is the constraint factor (= 1.363 for ASTM tup); and β is a constant depending on the yielding criterion, β = 2 for Tresca criterion. The calculated values for maximum stress are summarized in Table 4.…The results of crack energy (Tables 2 and 3) were in accordance with maximum stresses (Table 4). For the steels with ultra-low N content, the significant increment in maximum stress was observed for the tempartures ranging from -20°C to 25°C (Table 4). Opposite, for the 12CrTaNb0.03N steel, the TMP provided the increase in maximum stress at lower temperatures, only (Table 4).”
Point 4: The grain size for the three materials after CHT and TMP should be replenished.
Response 4: These data are represented in Tables 7-9.
Point 5: The creep properties are the most important properties for the materials, thus at least the creep strengths at high temperature after CHT and TMP should be replenished in the paper. In that case, readers can evaluatthe effectiveness of composition modification and the new heat treatment method.
Response 5: The creep properties require a lot of time and are in progress now. Authors will publish the creep results in the future works.
Reviewer 2 Report
1. English should be checked carefully. There are many misspellings. Some terms are peculiar, eg. "optic metallography".
2. If I understand it well the fraction of VX particles was determined using dark field images. Carbon extraction replicas were used for this. Orientation of VX particles on replicas is random. I think dark field images cannot reveal all VX particles in the area of interest. Furthermore, the lattice parameter of (Ta,Nb)X particles is close to that of VX particles. How can you avoid confusing these particles with different composition in dark field images? Why did you not use X-ray mapping? All vanadium rich particles in the area of interest would be identified easily.
3. In the equation (1) you calculate the volume as As (defined as "square of TEM image" - probably TEM image area) multiplied by the thickness of the carbon film. This calculation underestimates the real volume of sample corresponding to extracted particles on replica. Usually, the average diameter of extracted particles is used for calculations of the extraction volume.
4. I recommend to use the term "lath width" instead of "lath size", see Table 6.
5. In Tables 6-8 you mix experimental results and values calculated by using Thermocalc. Are you sure that fractions of minor phases in your samples comply with thermodynamic equilibrium?
Author Response
Point 1: English should be checked carefully. There are many misspellings. Some terms are peculiar, eg. "optic metallography".
Response 1: Done.
Point 2: If I understand it well the fraction of VX particles was determined using dark field images. Carbon extraction replicas were used for this. Orientation of VX particles on replicas is random. I think dark field images cannot reveal all VX particles in the area of interest. Furthermore, the lattice parameter of (Ta,Nb)X particles is close to that of VX particles. How can you avoid confusing these particles with different composition in dark field images? Why did you not use X-ray mapping? All vanadium rich particles in the area of interest would be identified easily.
Response 2: Authors agree that estimation of volume fraction of VX via dark field in TEM has strict limits. First, the chemical compositions of the particles, which was reflected in dark field, were checked using EDS; and only particles enriched in V were counted. Second, it is good idea to use X-ray mapping, but STEM regime gives the low quality of images with high magnification.
Point 3. In the equation (1) you calculate the volume as As (defined as "square of TEM image" - probably TEM image area) multiplied by the thickness of the carbon film. This calculation underestimates the real volume of sample corresponding to extracted particles on replica. Usually, the average diameter of extracted particles is used for calculations of the extraction volume.
Response 3: The Eq. 1 was corrected (p. 11). The text was corrected in accordance with new calculation: “The volume fraction of V-rich MX was 0.04%.” (p.11), “The volume fraction was also estimated using Eq. (1) and comprised 0.03%.” (p. 13). “The last method provided the volume fraction of VX phase of 0.08% regardless of the treatment.” (p. 15).
Point 4. I recommend to use the term "lath width" instead of "lath size", see Table 6.
Response 4: Done.
Point 5. In Tables 6-8 you mix experimental results and values calculated by using Thermocalc. Are you sure that fractions of minor phases in your samples comply with thermodynamic equilibrium?
Response 5: The estimation of volume fraction of minor phases is complicated task. Using Thermo-Calc software helps to find the reference point, because the calculation using TEM data also may be far from reality. Sure, the real fraction of minor phases can differ from thermodynamic equilibrium. In the present study, these values have informative character.
Reviewer 3 Report
Through their the papers published so far the authors demonstrate the continuity of research in the field of Thermo-Mechanical Processing, one gets the impression that this manuscript is only a variant of the already published paper Effect of the Thermo-Mechanical Processing on the Impact Toughness of a 12% Cr Martensitic Steel with Co, Cu,W, Mo and Ta Doping published in the journal Metals in December 2021.
In the manuscript the authors cited 31 literature references, of which 25 are the same literature references as in the above-mentioned paper. The first 13 citations appear in the same order in both manuscript and the already published paper. The authors cite papers [1-8], [9-12] in the same way, even though they carried out a different research process. The work is conceived in exactly the same way , and one has the impression that this paper was created by replacing some parts of the already published paper mentioned above. Excessive similarity with the already published paper reduces the quality of the manuscript and leads to the conclusion that the manuscript was written in haste without a scientific approach. Considering that a lot of tests and measurements were carried out in the manuscript, it is suggested that the authors revise the text, primarily the introduction, conclusion and literature. Authors should, when amending the manuscript to demonstrate a scientific approach to research and writing. Verbatim repetition and copying of the form and text are not permissible in a scientific paper.
Author Response
Point 1: Through their the papers published so far the authors demonstrate the continuity of research in the field of Thermo-Mechanical Processing, one gets the impression that this manuscript is only a variant of the already published paper Effect of the Thermo-Mechanical Processing on the Impact Toughness of a 12% Cr Martensitic Steel with Co, Cu,W, Mo and Ta Doping published in the journal Metals in December 2021.
In the manuscript the authors cited 31 literature references, of which 25 are the same literature references as in the above-mentioned paper. The first 13 citations appear in the same order in both manuscript and the already published paper. The authors cite papers [1-8], [9-12] in the same way, even though they carried out a different research process. The work is conceived in exactly the same way , and one has the impression that this paper was created by replacing some parts of the already published paper mentioned above. Excessive similarity with the already published paper reduces the quality of the manuscript and leads to the conclusion that the manuscript was written in haste without a scientific approach. Considering that a lot of tests and measurements were carried out in the manuscript, it is suggested that the authors revise the text, primarily the introduction, conclusion and literature. Authors should, when amending the manuscript to demonstrate a scientific approach to research and writing. Verbatim repetition and copying of the form and text are not permissible in a scientific paper.
Response 1: The paper published in the journal Metals in December 2021 was devoted to the thermo-mechanical processing of the 12% Cr steel. We showed the effect of delta-ferrite on the impact tougnhess and suggested the methods how to avoid the delta-ferrite formation. However, such thermo-mechanical processing provided a little decrease in the ductile-brittle transition temperature (DBTT) from 70°C (after conventional heat treatment) to 65°C.
The next step of our scientific approach was the modification of the thermo-mechanical processing via decreasing the temperature of homogenization annealing and changing the tempering temperature (these results were sent to other journal for publication). This led to the significant improving impact toughness and shifting DBTT to the side of lower temperatures on 30 K. We determined the optimal regime of the thermo-mechanical processing for further investigations.
The next step of our scientific approach was to check the application of the modified thermo-mechanical processing for other 12% Cr steels with different alloying. This was presented in the present study. Three steels were subjected to the optimal modified thermo-mechanical processing with aim to improve their impact toughness. The obtained results with discussion were reported in the present paper.
These three steps of scientific way are consistent and related to each others, but they have the significant differences in the goals, materials and implementation. Sure, for analysis of impact toughness and structural parameters, we used the standard instrumental tools to have an opportunity to compare results without additional external factors. Moreover, the literature references in both present paper and paper published in Metal 2021 were similar because of the similarity of topic as well as the fact that the references didn't lose their actuality for one year. The Introduction demonstrates the key points about 12% Cr steels and opens the reasons of low impact toughness. Moreover, in Introduction, we noticed the paper published in Metal 2021 as the initial investigation and concluded the advantages and disadvantages of the previous thermo-mechanical processing. The aim of the present paper is based on elimination of disadvantages of the previous thermo-mechanical processing and expands the application of modified processing for other steels. The conclusion of the present study reflects the main results obtained for different steels. This indicates that our scientific approach has direct, consistent and successive character: research → modification → application to other materials.
Reviewer 4 Report
The present work investigated the Effect of thermo-mechanical processing on the impact toughness of the 12% Сr steels with low N and high B contents mainly through decreasing delta-ferrite. This paper is well written with detailed experimental data, and before it can be accepted for publication in Materials, the following comments should be considered:
1. In the introduction section, the reason of high Cr and high B in the novel steels were explained, however, the role of low N was lack, and is suggested to be completed.
2. Since the structure of the carbides, e.g. M23C6, and VX,NbX etc. were characterized by SAED, and the delta ferrite is import to the improvement of the investigated steels subjected to TMP, the structure of delta ferrite is recommended to be characterized by SAED in TEM.
3. A few writing and grammar problems need further improvement, for example, the “favourably” in the sentence of “The precipitation of fine MX with high volume fraction is favourably”(on page 15, paragraph 6) should be”favourble”, etc.
Author Response
Point 1: In the introduction section, the reason of high Cr and high B in the novel steels were explained, however, the role of low N was lack, and is suggested to be completed.
Response 1: The text was added: “The effective method to prevent the formation of Z-phase in 10-12% Cr steels during long-term creep or ageing is a decrease in nitrogen content to 0.01 wt.% or less [7, 17, 18]. This eliminates both V-rich MX and Z-phase particles [17]. The decrease in N content gives an opportunity to increase in B content without the formation of BN nitrides at high temperatures [1]. High B content positively effects on the dispersion of M23C6 carbides decreasing their mean size that improves the creep resistance [1].” (p. 1)
Point 2: Since the structure of the carbides, e.g. M23C6, and VX,NbX etc. were characterized by SAED, and the delta ferrite is import to the improvement of the investigated steels subjected to TMP, the structure of delta ferrite is recommended to be characterized by SAED in TEM.
Response 2: Morphology of delta-ferrite allows to distinguish this phase on the images obtained by optical metallography, SEM and TEM. SAED in TEM for delta-ferrite doesn’t provide the additional information about this phase. Delta-ferrite and alpha-ferrite in the form of tempered lath structure have the same lattices (BCC) with similar lattice parameter, i.e using SAED in TEM doesn’t help to separate these phases from each others.
Point 3: A few writing and grammar problems need further improvement, for example, the “favourably” in the sentence of “The precipitation of fine MX with high volume fraction is favourably”(on page 15, paragraph 6) should be”favourble”, etc.
Response 3: Done.
Round 2
Reviewer 1 Report
The authors have not revised the paper according to the reviewer's comments or correctly reply them.
Author Response
Point 1. The authors have not revised the paper according to the reviewer's comments or correctly reply them.
Response 1: Authors were surprised the Reviewer’s comment. Authors added the information about the method of melting, the amount of oxygen and methods of analysis according to Comments 1 and 2 (p. 2). In accordance with Comment 3, we calculated the maximum stress via maximal load using "load vs. deflection” curves and added the Table 4 with some comments for these results (p. 6). Data on grain size for Comment 4 were represented in Table 7-9 in the second column (PAG size) (p. 11, 13, 15). Now creep data for Comment 5 are not ready to be published, because some creep tests under different applied stresses are in progress and not finished. Analysis of unfinished creep data is not correct: (1) the creep tests under the high applied stress leads to overestimation of long-term creep, and (2) the creep tests under the low applied stress have not reached even the steady-state creep stage. This indicates that an addition of creep data to this paper is premature and inadvisable.
Reviewer 2 Report
Thank you for your responses. Quantitative metallography is a difficult task.
Author Response
Thank you for your Comments. We will try to use X-ray mapping in future investigations.
Reviewer 3 Report
In the revised version of the manuscript the authors did not consider the comments.
The authors just added three references. They did not refer to the reference number 40 anywhere in the revised version.
In the first version of the manuscript, the authors state:
It is known that good creep resistance is accompanied with the large mean size of prior austenite grains and relatively fine M23C6 carbides densely located along lath boundaries [37]
and quote the paper [37] Fedoseeva, A.; Nikitin, I.; Tkachev, E.; Mishnev, R.; Dudova, N.; Kaibyshev, R. Effect of alloying on the nucleation and growth of Laves phase in the 9–10%Cr-3%Co martensitic steels during creep. Metals. 2021, 11, 60. https://doi.org/10.3390/met11010060.
In the revised version for the same statement:
It is known that good creep resistance is accompanied with the large mean size of prior austenite grains and relatively fine M23C6 carbides densely located along lath boundaries [39]
they quote the paper: [39] Dudova, N.; Mishnev, R.; Kaibyshev, R. Creep behavior of a 10%Cr heat-resistant martensitic steel with low nitrogen and high boron contents at 650 °C. Mater. Sci. Eng. A. 2019, 766, 138353. http://doi.org/10.1016/j.msea.2019.138353.
The revised manuscript confirms my assumptions that I wrote in the review. The authors spent more time convincing us that the paper was a continuation of the research than they did on writing the manuscript.
Author Response
Point 1: In the revised version of the manuscript the authors did not consider the comments.
The authors just added three references. They did not refer to the reference number 40 anywhere in the revised version.
In the first version of the manuscript, the authors state:
It is known that good creep resistance is accompanied with the large mean size of prior austenite grains and relatively fine M23C6 carbides densely located along lath boundaries [37]
and quote the paper [37] Fedoseeva, A.; Nikitin, I.; Tkachev, E.; Mishnev, R.; Dudova, N.; Kaibyshev, R. Effect of alloying on the nucleation and growth of Laves phase in the 9–10%Cr-3%Co martensitic steels during creep. Metals. 2021, 11, 60. https://doi.org/10.3390/met11010060.
In the revised version for the same statement:
It is known that good creep resistance is accompanied with the large mean size of prior austenite grains and relatively fine M23C6 carbides densely located along lath boundaries [39]
they quote the paper: [39] Dudova, N.; Mishnev, R.; Kaibyshev, R. Creep behavior of a 10%Cr heat-resistant martensitic steel with low nitrogen and high boron contents at 650 °C. Mater. Sci. Eng. A. 2019, 766, 138353. http://doi.org/10.1016/j.msea.2019.138353.
The revised manuscript confirms my assumptions that I wrote in the review. The authors spent more time convincing us that the paper was a continuation of the research than they did on writing the manuscript.
Response 1:
1. Mention of missed Ref [40] was corrected as Ref [41] “It is known that good creep resistance is accompanied with the large mean size of prior austenite grains and relatively fine M23C6 carbides densely located along lath boundaries [41]”. This Ref [41] corresponds to Ref [37] in initial manuscript. It was technical error related to addition of new references.
2. In Introduction, the changes were added as follows: “As it was mentioned in [33], the significant increment in entire Charpy energy curve for the 12% Cr steel via the thermo-mechanical processing was accompanied with the structural changes in (a) the delta-ferrite amount, (b) the dispersion of M23C6 carbides and (c) an additional precipitation of nanoscale MX carbonitrides. However, such thermo-mechanical processing provides a little decrease in the DBTT from 70°C (after conventional heat treatment) to 65°C [33]. This led to the modification of the thermo-mechanical processing via decreasing the temperature of homogenization annealing from 1200°C to 1150°C as it was in conventional heat treatment. Such modification provides the significant improving impact toughness and shifting DBTT to the side of lower temperatures by 20 K [34]. However, the application of such modified thermo-mechanical processing for other 12% Cr steels with different alloying is open question. The present research is consistent development of previous works directed to improving the impact toughness of the 12% Cr steels. So, the aim of the present research is verification of the positive effect of modified thermo-mechanical processing for other 12% Cr steels with different alloying containing low N and high B additives.” (p. 2).
3. References [1], [3-5] and [9] were replaced:
[1] Kern, T.-U.; Staubli, M.; Scarlin, B. The European efforts in material development for 650°C USC power plants—COST522. ISIJ Int. 2002, 42, 1515-1519. http://doi.org/10.2355/isijinternational.42.1515.
[3] Abe, F.; Tabuchi, M.; Tsukamoto, S. Alloy Design of MARBN for Boiler and Turbine Applications at 650°C. Mater. at High Temp. 2021, 38, 306-321. http://doi.org/10.1080/09603409.2021.1963393.
[4] Maruyama, K.; Sekido, N.; Yoshimi, K. Changes in strengthening mechanisms in creep of 9Cr-1.8W-0.5Mo-VNb steel tested over wide ranges of creep conditions. Int. J. Press. Vessel. Pip. 2021, 190, 104312. https://doi.org/10.1016/j.ijpvp.2021.104312.
[5] Sawada, K.; Kimura, K.; Abe, F.; Taniuchi, Y.; Sekido, K.; Nojima, T.; Ohba, T.; Kushima, H.; Miyazaki, H.; Hongo, H.; Watanabe, T. Catalog of NIMS creep data sheets. Sci. Technol. Adv. Mater. 2019, 20, 1131-1149. https://doi.org/10.1080/14686996.2019.1697616.
[9] Sklenička, V.; Kuchařová, K.; Kvapilová, M.; Kloc, L.; Dvořák, J.; Král, P. High-temperature creep tests of two creep-resistant materials at constant stress and constant load. KEM 2019, 827, 246–251. https://doi.org/10.4028/www.scientific.net/kem.827.246.
The Ref. [34] was added:
[34] Fedoseeva, A.; Kaibyshev, R. Impact toughness of a 12% Cr martensitic steel: conventional heat treatment vs. thermo-mechanical processing. Mater. Lett. 2022, submitted.